# Tumor-Specific miRNA Signatures in Combination with CA19-9 for Liquid Biopsy-Based Detection of PDAC

**DOI:** 10.3390/ijms222413621

**Published:** 2021-12-19

**Authors:** Min Woo Kim, Hani Koh, Jee Ye Kim, Suji Lee, Hyojung Lee, Young Kim, Ho Kyoung Hwang, Seung Il Kim

**Affiliations:** 1Division of Breast Surgery, Department of Surgery, Yonsei University College of Medicine, Seoul 03722, Korea; minwookim@yuhs.ac (M.W.K.); JEEYE0531@yuhs.ac (J.Y.K.); SOOJEE2@yuhs.ac (S.L.); HJLEE709@yuhs.ac (H.L.); youngkim@ens-h.com (Y.K.); 2Division of Hepatobiliary and Pancreatic Surgery, Department of Surgery, Yonsei University College of Medicine, Seoul 03722, Korea; HANIKOH@yuhs.ac

**Keywords:** pancreatic cancer, diagnosis, liquid biopsy, extracellular vesicles, microRNA, CA19-9

## Abstract

Pancreatic ductal adenocarcinoma (PDAC) is considered one of the most aggressive malignancies and has high mortality and poor survival rates. Therefore, there is an urgent need to discover non-invasive biomarkers for early detection before PDAC reaches the incurable stage. We hypothesized that liquid biopsy of PDAC-derived extracellular vesicles (PDEs) containing abundant microRNAs (miRNAs) could be used for early diagnosis of PDAC because they can be selectively enriched and because they are biologically stable. We isolated PDEs by immunocapture using magnetic beads, and we identified 13 miRNA candidates in 20 pancreatic cancer patients and 20 normal controls. We found that expression of five miRNAs, including miR-10b, miR-16, miR-155, miR-429, and miR-1290, was markedly higher in PDEs. Furthermore, the miRNA signatures along with serum carbohydrate antigen 19-9 (CA19-9) were optimized by logistic regression, and the miRNA signature and CA19-9 combination markers (CMs) were effective at differentiating PDAC patients from normal controls. As a result, the CMs represented a high sensitivity (AUC, 0.964; sensitivity, 100%; specificity, 80%) and a high specificity (AUC, 0.962; sensitivity, 85.71%; specificity, 100%). These findings suggest that five miRNAs expressed in PDEs and CA19-9 are valuable biomarkers for screening and diagnosis of pancreatic cancer by liquid biopsy.

## 1. Introduction

Pancreatic ductal adenocarcinoma (PDAC) is the most prevalent type of pancreatic cancer and is the leading cause of cancer-related death [1]. PDAC is very silent and has occasionally already reached the advanced disease with distant metastasis stage upon diagnosis. Due to the asymptomatic nature of PDAC, it is usually detected by chance during medical checkups for other diseases [2]. No doubt, tissue biopsy following imaging techniques, such as computed tomography, magnetic resonance imaging, and ultrasonography [3,4,5], is a definitive method with a high diagnostic accuracy; however, it is sometimes unavailable for analysis, hindering diagnosis [6]. It can be limited when a tumor mass is insufficient or in an inaccessible location, for instance, at a distant organ site where metastasis occurred. Moreover, repetitive imaging and tissue collection increase burden to the patients.

Today, liquid biopsy is a potential diagnostic modality that is complementary rather than an alternative to the conventional diagnostic methods. Liquid biopsy may provide transcriptomic and proteomic information on the spatial and temporal heterogeneity of a tumor [7,8]. A single tissue biopsy sample may be biased and controversial, while liquid biopsy is relatively non-invasive, easy to repeat, and provides overall information from tumor. Molecular information obtained by liquid biopsy can provide a surgeon with more detail to guide treatment and plan surgery. In this way, screening, diagnosis, prognosis, and treatment for cancer patients may change dramatically in the near future through liquid biopsy.

Existing blood tests to look for clues of cancer through liquid biopsy include the serum carbohydrate antigen 19-9 (CA19-9) circulating biomarker that can be utilized in diagnosis and monitor pancreatic cancer. However, elevated CA19-9 has also been found in some patients with benign pancreatobiliary diseases, suggesting that CA19-9 does not have high cancer marker specificity [9,10]. In addition to false positives, the other limitations of liquid biopsy are that circulating biomarkers are usually present at low concentrations in blood and can degrade quickly [11]. Recently, extensive studies confirmed that several microRNAs (miRNAs) are dysregulated in PDAC patients. Thus, serum miRNA expression patterns may have the potential to identify various stages of pancreatic cancer from early cancer to metastatic cancer [12]. The transfer of miRNAs to adjacent cells is induced primarily by extracellular vesicles (EVs), which load valuable markers (protein, mRNA, miRNA, and DNA) that reflect the properties of their parental cells [13]. Evs are abundant in circulating blood, can circulate for a long time, and protect molecular information derived from parental cells. Therefore, tumor-derived Evs have been proposed as potential biomarkers for cancer diagnosis.

As illustrated in Figure 1, this study was aimed to identify dysregulated miRNAs in PDAC-derived EVs (PDEs) using magnetic bead-based immunocapture for EV isolation. We expected that miRNAs relevant to tumorigenesis are found in primary tumors and enriched in tumor-derived EVs, distinct to other EVs that originated from non-cancerous cells [14]. Previously, we demonstrated that analysis of miRNA expression in EVs can diagnose cancer with high sensitivity and specificity [15]. In this study, we compared miRNA expression patterns in tumor tissue, total EVs (TEVs), and PDEs to identify a miRNA expression profile specific to PDEs. Then, we integrated the PDE-specific miRNAs with conventional CA19-9 analysis to determine which marker combination most effectively diagnosed PDAC.

## 2. Results

### 2.1. Characterization of EVs from Pancreatic Cancer Cell Lines

To verify that EVs are highly expressing candidate surface proteins that have been identified as cancer-associated markers in most cancer types, including PDAC [16,17,18], fluorescent intensities were measured by flow cytometry. EVs in the supernatants of five different parental cell lines were isolated using microbeads conjugated with antibodies against integrin alpha 2 (ITGA2), integrin alpha V (ITGAV), epithelial cell adhesion molecule (EpCAM), and glypican−1 (GPC1). Compared with integrin marker levels in the normal pancreatic duct cell line HPNE, integrin marker levels trended higher in AsPC−1, BxPC−3, and Mia PaCa−2 cells and EVs; however, the levels of the EpCAM and GPC1 showed inconsistent patterns between cells and EVs (Figure 2A). In particular, GPC1, which was expected to be expressed low in pancreatic cancer (Appendix A), was actually highly expressed in EVs. It indicated that surface proteomes of EVs and cells might show some discrepancy. For better characterization of the immunocaptured EVs, we made a scatter plot of the protein expression patterns (Figure 2B). Although no statistically significant correlation was observed between cells and EVs, GPC1 was expressed highly in EVs (quadrant 1) and integrin markers were expressed highly in both cells and EVs (quadrant 2). Additionally, relative expression of the surface markers in cells and EVs is represented in heatmaps (Figure 2C). The mean fold changes of ITGA2, ITGAV, EpCAM, and GPC1 expression were 37.1, 14.3, 36.1, and 1.8 in cells and 72.0, 66.4, 11.5, and 51.9 in EVs, respectively. Based on these results, ITGA2, ITGAV, and GPC1 were used to isolate tumor-derived EVs in subsequent experiments.

### 2.2. Immunocapture Using Magnetic Beads for Tumor Cell-Derived EV Isolation

To establish a high-throughput EV isolation method applicable to clinical analysis, we utilized magnetic beads that can selectively enrich for EVs containing ITGA2, ITGAV, and GPC1. First, we tested our system with EV samples isolated from pancreatic cancer and normal cell lines. Nanoparticle tracking analysis (NTA) allowed EV concentrations to be estimated up to 1 × 10^10^ EVs per mL with particle sizes ranging from 50–300 nm in diameter (Figure 3A). The mean particle sizes of EVs isolated from culture supernatants of the five cell lines were 185.0 nm for HPNE, 201.5 nm for AsPC−1, 149.4 nm for BxPC−3, 227.5 nm for Mia PaCa−2, and 178.9 nm for PANC−1 (Figure 3B). Scanning electron microscopy (SEM) was used to confirm the presence of tumor cell-derived EVs along with their morphologies and sizes (Figure 3C). An equivalent concentration (10^9^ particles/mL) of EVs in phosphate buffered saline (PBS) was incubated with magnetic beads to isolate EVs by immunoaffinity (Appendix A). Similar to the NTA results, the EVs isolated with the magnetic beads were <250 nm in diameter and the majority were spherical. Although the aggregation of EVs may be a concern, we regarded it as a minor morphological change that only appeared in in vitro experiments when high-speed centrifugation was used to prepare a highly concentrated EV solution. The number of HPNE normal cell-derived EVs per bead was relatively low because ITGA2, ITGAV, and GPC1 are not expressed highly enough in the HPNE-derived EVs to reach saturation upon isolation. Because tumor-derived EVs could be surrounded by large amounts of other EVs released from normal tissues in biological environments, our magnetic bead-based EV isolation method is required to separated them from other contaminants.

### 2.3. Identification of Differentially Expressed miRNA Candidates

Based on multiple experimental and bioinformatic databases, we selected thirteen miRNAs for RT-qPCR expression analysis in four pancreatic cancer cell lines (Figure 4A) and tumor cell-derived EVs (Figure 4B) normalized to HPNE cells and EVs. Of the miRNAs analyzed, eight (miR-10b, miR-16, miR-21, miR-96, miR-103, miR-155, miR-429, and miR-1290) were upregulated and four (miR-16, miR-4732, miR-4644, and miR-138) were downregulated in EVs (Figure 4C). The overall miRNA expression patterns between parental cells and their released EVs were similar; however, miR-9 and miR-3976 were expressed highly in the parental cells but not in EVs. We identified seven miRNAs that were upregulated >1.5−fold in EVs (miR-10b, miR-21, miR-96, miR-103, miR-155, miR-429, and miR-1290). These miRNAs are candidate liquid biopsy PDE biomarkers for diagnosis of pancreatic cancer in the clinical setting.

To obtain accurate, clinically relevant, and reliable miRNA RT-qPCR data, the stability of each miRNA was determined. Because no standard internal controls for the normalization of miRNA within EVs have been established, we used miRNAs that are stably expressed in plasma as reference miRNAs, and we validated their stability in 20 PDAC patients and 20 cholecystitis (CL) controls. The appropriate reference miRNA genes were identified for each sample Cq value using statistical algorithms, including BestKeeper, NormFinder, the Delta Ct method, and GeNorm (Figure 5A). A comprehensive ranking of the reference genes was generated by RefFinder, which considers the results generated by four algorithms. Based on this ranking, miR-16 was the most stable reference miRNA gene in PDEs from 40 plasma samples (Figure 5B) and RNU6B was the most stable miRNA gene in 20 tumor tissue and 20 adjacent tissue samples (Appendix A). These results were consistent with other reference gene validation studies, indicating that the experimental setting of our gene stability analysis was credible.

### 2.4. Comparative Analysis of miRNA Candidates in Tissue and Plasma

Based on our in vitro cell-EV analysis, we speculated that miRNA expression may differ between PDEs and total EVs (TEVs) or tumor tissue from patients. To analyze expression patterns in PDAC patients, expression of the seven candidate miRNAs in PDEs, TEVs, and tissue were assessed (Figure 6A). Expression of five of the seven candidate miRNAs (miR-10b, miR-21, miR-155, miR-429, and miR-1290) was upregulated in PDEs. No differences in expression were found for miR-96, regardless of the miRNA separation method. miR-103 expression was upregulated in tumor tissue only. Analysis of expression in TEVs showed no significant differences. In conclusion, expression of miR-10b, miR-21, miR-155, miR-429, and miR-1290 was 1.83-, 1.91-, 2.00-, 3.02-, and 3.61-fold higher in PDEs compared to EVs from non-PDAC patients (Figure 6B and Appendix A). Therefore, these five miRNAs were further validated for multi-panel analysis.

### 2.5. Prediction Performance of Combined Multiple Biomarkers

To evaluate the diagnostic power of each miRNA in PDEs alone, we performed receiver operating characteristic (ROC) curve analysis with the same patient group and set the area under ROC curve (AUC) cutoff for good diagnostic value as >0.65. Of the 13 identified miRNAs, 5 miRNAs (miR-10b, miR-21, miR-155, miR-429, and miR-1290) met the AUC criteria. Therefore, we evaluated them further for their potential as diagnostic biomarkers for pancreatic cancer. The AUC values for miR-10b, miR-21, miR-155, miR-429, and miR-1290 were 0.693 (95% confidence interval (CI), 0.529–0.827), 0.771 (95% CI, 0.614–0.888), 0.843 (95% CI, 0.695–0.937), 0.867 (95% CI, 0.529–0.827), and 0.786 (95% CI, 0.630–0.898), respectively (Figure 7A).

Based on clinical records of conventional protein biomarkers, such as carcinoembryonic antigen (CEA; cutoff, 5 ng/mL; AUC, 0.771; sensitivity, 19.0%; specificity, 95.0%) and CA19-9 (cutoff, 38.5 U/mL; AUC, 0.780; sensitivity, 47.62%; specificity, 100.0%), the significance of miRNA biomarkers in clinical diagnosis is not yet clear, and this encouraged us to combine putative biomarkers and evaluate the combinations for their accuracy in diagnosing PDAC. We used logistic regression to evaluate the performance of miRNA multi-panels. As summarized in Appendix A, we selected four optimized miRNA panels (panel 1: miR-21, miR-155, miR-429, and miR-1290; panel 2: miR-10b, miR-21, miR-155, and miR-1290; panel 3: miR-10b, miR-155, miR-429, and miR-1290; panel 4: miR-10b, miR-21, miR-155, miR-429, and miR-1290) and assembled them with the CA19-9 protein biomarker. Each combination marker (CM 1–4), which consisted of the miRNA panel and CA19-9, showed a higher risk of pancreatic cancer algorithm (ROPCA) score for PDAC patients than the normal control, and this indicates that the markers can identify increased PDAC risk (Figure 7B). The CM with the most sensitive diagnostic power was miRNA panel 3 plus CA19-9 (AUC, 0.964; sensitivity, 100%; specificity, 80%) and the most specific CM was miRNA panel 1 plus CA19-9 (AUC, 0.962; sensitivity, 85.71%; specificity, 100%), (Figure 7C).

## 3. Discussion

Although there have been recent advancements in the diagnosis of cancer by liquid biopsy, pancreatic cancer is still confirmed clinically through traditional imaging methods followed by tissue biopsy. However, the traditional diagnostic methods are invasive and difficult to repeat; thus, they limit cancer diagnosis and follow-up, and development of a new diagnostic modality is crucial. Recently, a steadily increasing number of reports have shown that tumor-associated miRNAs can help accurately diagnose cancer [19,20]. Upregulation of certain miRNAs in pancreatic cancer tissue compared to normal pancreatic tissue was shown in multiple pancreatic cancer-associated miRNA profiling studies [21,22,23,24,25]. In this study, we validated the feasibility of liquid biopsy using EVs because liquid biopsy detects alterations in blood rather than in the tumor itself. Indeed, a major limitation of using tumor tissue for diagnosis is that it can be critically affected by clinical heterogeneity [26]. For these reasons, we analyzed expression of a panel of candidate miRNAs in PDEs to evaluate their ability to diagnose PDAC.

To the best of our knowledge, only a few cancer-related studies investigated expression of miRNA in various source samples. In this study, to identify a method suitable for clinical use, we compared miRNA expression patterns in three different miRNA sources, tumor tissue, TEVs, and PDEs, that associate strongly with pancreatic cancer. In fact, considerable heterogeneity can be observed in biological environments using EVs because EVs are released from almost all cell types. Currently, several commercial EV isolation kits based on PEG precipitation (Total Exosome Isolation kit and ExoQuick) and ultracentrifugation (UC) are used for TEV isolation [27]. TEV isolation allows for higher yields compared to the immunocapture-based EV isolation technologies; however, TEV cannot capture cancer heterogeneity or reflect tumor properties. The presence of contaminants and other vesicles prevent the use of commercial isolation methods. Therefore, we differentiated PDEs from TEVs using the candidate cancer surface markers ITGA2, ITGAV, and GPC1.

Pearson correlation analysis showed that higher correlations were observed between miRNA expression levels in PDEs and tumor tissue than between miRNA expression levels in TEVs and tumor tissue (Appendix A). Fold changes of miRNA expression showed that miR-10b, miR-21, miR-155, miR-429, and miR-1290 were highly expressed in PDEs, whereas only two of the five miRNAs (miR-21 and miR-155) were significantly elevated in tumor tissues. In addition, none of the miRNAs showed significantly increased expression in TEVs; thus, expression of miRNAs in TEVs was of no diagnostic value in our experiments. These results indicate that PDEs can reflect cancer heterogeneity and that the process of packing miRNA into EVs could be selective. A large majority of biomarkers may be shared between EVs and their parental cells, but direct application of tissue-associated markers remains controversial. In summary, the more selectively that PDEs are separated from TEVs, the more accurately the biomarkers within EVs are profiled.

Here, we found synergistic effects when a panel of miRNAs and the serum CA19-9 biomarker were combined to evaluate PDAC patients. Notably, the combination of miR-21, miR-155, miR-429, miR-1290, and CA19-9 showed the highest specificity. The combination of miR-10b, miR-155, miR-429, miR-1290, and CA19-9 had the highest sensitivity. The value of the miRNA panels as diagnostic markers was confirmed by ROC curve analysis, and each miRNA panel had higher diagnostic efficacy than the traditional biomarkers CEA and CA19-9 (Appendix A). These results show that expression of miR-10b, miR-21, miR-155, miR-429, and miR-1290 in PDEs may be an indicator of pancreatic cancer risk. As known oncogenic miRNAs, these miRNAs promote cancer initiation and progression, but the relationships between them and pancreatic cancer is not yet clear. Further functional evaluation of the miRNAs in pancreatic cancer is required.

In this study, we found that a cocktail of three protein markers (ITGA2, ITGAV, and GPC1) allowed for isolation of PDEs. However, identification of highly expressed miRNAs in PDEs was restricted by these targets. Thus, efforts to elucidate the surface proteome of PDEs and to compare different surface markers may allow for more specific identification of miRNA candidates. For instance, Castillo et al. demonstrated a panel of PDAC related EV surface markers selectively enriched within EVs derived from 13 human PDAC cell lines. These candidate markers (CLDN4, EPCAM, CD151, LGALS3BP, HIST2H2BE, and HIST2H2BF) effectively isolated tumor-derived EVs in clinical samples [28]. In addition, we will require expanded patient cohorts to show the potential diagnostic value of PDEs. Owing to the nature of an exploratory study with small sample size, our results may not generalize to the larger population. Instead, in the present study, we aimed at comparing tissues (*n* = 20 + 20, tumor and adjacent tissue), TEVs (*n* = 20 + 20, PDAC and CL), and PDEs (*n* = 20 + 20, PDAC and CL) to demonstrate that PDEs can well reflect cancer heterogeneity and molecular profiles of tumor tissues. Future work will focus on validating the five miRNAs in a prospective study, which will be designed to screen enough patients to predict clinically interesting findings beyond early diagnosis. As further comparative studies are accumulated, the clinical value of EV-based liquid biopsy will increase.

## 4. Materials and Methods

### 4.1. Cell Lines and Cell Culture

The following four pancreatic cancer cell lines and the normal pancreas cell line were purchased from American Type Culture Collection (ATCC; Manassas, VA, USA): AsPC−1 (CRL−1682), BxPC−3 (CRL−1687), Mia PaCa−2 (CRL−1420), PANC−1 (CRL−1469), and *hTERT*-immortalized human pancreatic epithelial Nestin-expressing (HPNE; CRL−4023). All cell lines were cultured in complete growth medium as described in the handling information provided by ATCC. HPNE was immortalized by transduction with a retroviral expression vector (pBABEpuro) containing the *hTERT* gene [29].

### 4.2. Clinical Cohorts

In total, 40 subjects (20 cancer patients with PDAC and 20 non-cancer patients with CL) participated in this study. Informed consent for the use of tissue and matched blood samples for research purposes was obtained from all participants. Clinical samples were obtained from subjects who had visited Severance Hospital in South Korea in accordance with the guidelines of the independent Ethics Committee at the College of Medicine Yonsei University (IRB No. 4-2020-1292, approved on 4 January 2021). The cancer tissues and adjacent normal pancreas tissues were collected under the judgment of an experienced surgeon during surgical removal of pancreatic cancer tissue from patients with PDAC. The pre-operative plasma samples from same patients were collected before anesthesia. Criteria for subject eligibility for inclusion in the analysis included: (1) no chemotherapy or radiotherapy prior to blood and tissue specimen acquisition, (2) a confirmed pathologic diagnosis of PDAC for cancer cohort enrollment, and (3) a confirmed diagnosis of benign disease with non-cancerous conditions of the gallbladder such as gallstones. The clinical characteristics of subjects enrolled in this study are summarized in Appendix A.

### 4.3. EV Enrichment and Isolation

To release EVs from cell lines, each cell line was cultured to 70–80% confluency in a 150 mm cell culture dish with regular media for 24 h, and then the regular media was replaced with media containing 10% EV-depleted fetal bovine serum (FBS). Next, cells were incubated for another 72 h before supernatants were collected and centrifuged at 2000× *g* for 20 min to obtain EVs without dead cells and debris. Then, the EV solution was concentrated using a Macrosep Advance Centrifugal Device (100 K, Pall Corporation, Port Washington, NY, USA) and centrifuging at 2000× *g* for 2 h. The concentrated EVs were stored at −80 °C until further use. EV-depleted FBS was generated using ultracentrifugation at 120,000× *g* (SW28 rotor, Beckman Coulter, Brea, CA, USA) for 6 h at 4 °C followed by ultrafiltration of the cell culture supernatant using a 0.22 μm syringe filter [30].

In the experimental setting using clinical samples, EV isolation was categorized as total EVs isolation (TEVs) and PDAC-derived EVs (PDEs) isolation. TEVs were isolated from 200 μL of plasma using the commercial precipitation-based Total Exosomes Isolation kit (Invitrogen, Pleasanton, CA, USA), whereas PDEs were isolated using an affinity-based method. For the capture of PDEs, magnetic beads (Dynabeads™ M−270 Streptavidin, Thermo Fisher Scientific, Waltham, MA, USA) conjugated with biotinylated antibodies against ITGA2, ITGAV, and GPC1 were used. EpCAM, GPC1, ITGA2, and ITGAV were selected as candidate cancer surface markers based on sorting criteria such as “Predicted Membranous proteins”, “Detected in all cancer”, and “strongly/moderate expression in pancreatic cancer tissues” by approaching the Human Protein Atlas (HPA; www.proteinatlas.org, accessed on 15 January 2021; Appendix A). The HPA program is an open-access database that aims to map all human proteins by integrating various omics technologies.

### 4.4. Flow Cytometric Analysis of Surface Profiles

The surface profiles of cells and EVs bound to 15 μm microbeads (SVP 150-4, SPHERO™ Streptavidin Coated Particles, Spherotech Inc., Lake Forest, IL, USA) coated with antibodies targeting tumor-associated surface markers were analyzed via flow cytometry. The cell and EV samples were washed twice with ice-cold PBS with 1% bovine serum albumin and 0.1% sodium azide. Each sample was incubated for 30 min at 4 °C in the dark with one test dose of fluorescent-labeled antibodies, rinsed twice with FACS buffer to prevent excessive reactions, and then analyzed using a flow cytometer (FACS LSRFortessa system, Becton Dickinson, Franklin Lakes, NJ, USA). Antibodies were fluorescently labeled using the Alexa Fluor 488 Antibody Labeling Kit (A20181, Thermo Fisher Scientific, Waltham, MA, USA). PE-Cy7−labeled antibodies against the general EV marker CD63 (BD Biosciences, San Jose, CA, USA) were used in the EV analysis. Detailed information on antibodies is listed in Appendix A.

### 4.5. Physicochemical Properties of EVs

The concentrations and size distributions of EVs resuspended in PBS were measured by NTA with the NanoSight NS300 system (Malvern Panalytical, Malvern, UK). Analysis was performed using NTA 3.1 software with default settings according to the manufacturer’s software manual and the camera focus was adjusted to distinctly visualize EVs not exceeding a particle signal. To confirm attachment of pancreatic cancer-derived EVs to antibody-conjugated magnetic beads, specimens were observed with a field emission SEM (MERLIN, Carl Zeiss, Jena, Germany). Prior to observation, EVs were fixed for 24 h in Karnovsky’s fixative (2% glutaraldehyde and 2% paraformaldehyde in 0.1 M phosphate buffer, pH 7.4) and washed twice with 0.1 M phosphate buffer for 30 min. EVs were post–fixed with 1% OsO4 for 2 h, dehydrated using a gradually ascending ethanol series (50–100%) and a critical point dryer (CPD300, Leica Biosystems, Wetzlar, Germany), and coated with platinum using an ion sputter (ACE600, Leica Biosystems, Wetzlar, Germany).

### 4.6. Analysis of miRNA Expression Pattern by RT-qPCR

Total RNA in cell lines (AsPC−1, BxPC−3, Mia PaCa−2, PANC−1, and HPNE) and homogenized tissues (20 samples of paired cancer and adjacent normal tissues) was extracted using TRIzol reagent (Invitrogen, Pleasanton, CA, USA). The HPNE cell line and adjacent normal tissues were used as normal controls. Total RNA in TEVs and PDEs was extracted using a Total Exosome RNA and Protein Kit (Invitrogen) according to the manufacturer’s protocol. RNA purity and quantity were determined using a NanoDrop 3000 spectrophotometer (Thermo Fisher Scientific, Waltham, MA, USA). Extracted RNA was reverse transcribed using a TaqMan microRNA Reverse Transcription Kit (Thermo Fisher Scientific, Waltham, MA, USA). Differential expression levels of 13 miRNAs (miR-9, miR-10b, miR-16, miR-21, miR-96, miR-103, miR-138, miR-155, miR-429, miR-1290, miR-3976, miR-4644, and miR-4732) were measured by performing cDNA amplification reactions with a TaqMan Universal PCR Master Mix, No AmpErase UNG, 2x (Thermo Fisher Scientific, Waltham, MA, USA) and a TaqMan miRNA Assay Kit (Thermo Fisher Scientific, Waltham, MA, USA) in a CFX96 Real-time PCR system (Bio-Rad, Hercules, CA, USA). Normalization of miRNA expression levels was performed using RNU6B as the reference control for tissue miRNAs and miR-16 as the reference control for miRNAs in EVs. The experiments were performed in triplicate and the ΔΔCT method was used to determine relative expression.

### 4.7. Statistical Analysis

The applicability of the tested endogenous control miRNAs was evaluated using NormFinder, geNorm, BestKeeper, and RefFinder algorithms, which were developed to assess the stability of a normalizing gene [31,32,33,34]. The fold changes of miRNA expression in pancreatic cancer patients and normal controls were analyzed using two-tailed Student t-tests. The fold changes of miRNA expression were transformed to natural logs to reduce bias and to follow the practice that Moore et al. used when developing their risk of ovarian malignancy algorithm (ROMA) risk-prediction equations [35].

The ROC analysis of microRNA panels was performed using MedCalc’s Logistic regression analysis (v20.014, The MedCalc Software Ltd., Ostend, Belgium) on all pancreatic cancer patients versus all non-cancer controls. We used univariate ROC analysis on each panel to obtain the ROC curve, AUC, and standard error (SE) of the AUC in order to evaluate the diagnostic power of each microRNA biomarker combination. After performing a univariate ROC analysis on each combination of microRNA biomarkers, we chose the “Best” four microRNA panels with an AUC of 0.95 and the lowest SE of AUC. Then, in order to achieve distinguished diagnostic performance, four microRNA panels were combined with the CA19-9 protein biomarker. Then, in order to achieve distinguished diagnostic performance, four microRNA panels were combined with the CA19-9 protein biomarker. The Risk of Pancreatic Cancer Algorithm was calculated through both selected 4 microRNA panels and CA19-9. It was assessed for diagnostic ability by ROC analysis using the logistic regression method. Based on these results, we developed the ROPCA arbitrary unit, and the ROPCA score was utilized to predict pancreatic cancer.

## 5. Conclusions

To date, several miRNAs have been discovered that play important roles in pancreatic cancer development, progression, and metastasis. This study revealed dysregulated miRNA expression in PDAC patients; thus, miRNA expression in PDEs may be used as potential biomarkers for the diagnosis of PDAC. We found that PDAC patients exhibited higher expression of five miRNAs (miR-10b, miR-21, miR-155, miR-429, and miR-1290) in PDEs when compared to the normal control. In addition, the integrated analysis of miRNAs and serum CA19-9 was shown to be a better modality for pancreatic cancer diagnosis. Further evaluation using an expanded prospective cohort is necessary to determine the clinical value of miRNA expression in PDEs.

## Figures and Tables

**Figure 1 ijms-22-13621-f001:**
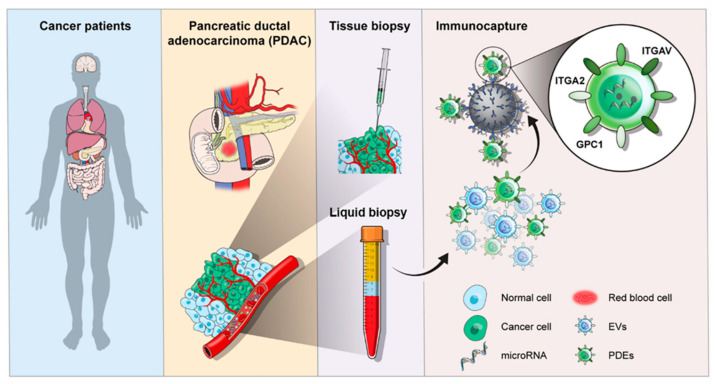
Schematic illustration of the clinical utility of EVs in liquid biopsy for the diagnosis of PDAC.

**Figure 2 ijms-22-13621-f002:**
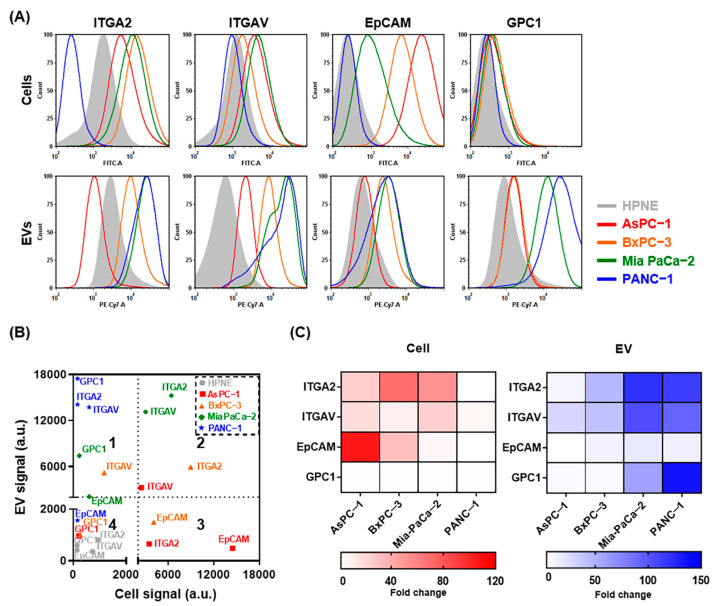
Characterization of pancreatic cancer-associated surface markers on the surface of cells and EVs. (**A**) Representative fluorescence-activated cell sorting (FACS) histograms of ITGA2, ITGAV, EpCAM, and GPC1 expression on cells (upper) and EVs (lower) using Alexa Fluor 488 and PE-Cy7−labeled antibodies. (**B**) Correlation between surface marker expression on cells (X-axis) vs. EVs (Y-axis). Dotted cutoff lines are shown at the mean fluorescence intensity (MFI) value of 2000. (**C**) Heat map of surface marker expression in pancreatic cancer cells (red) and EVs (blue) compared with the normal pancreas control cell line HPNE.

**Figure 3 ijms-22-13621-f003:**
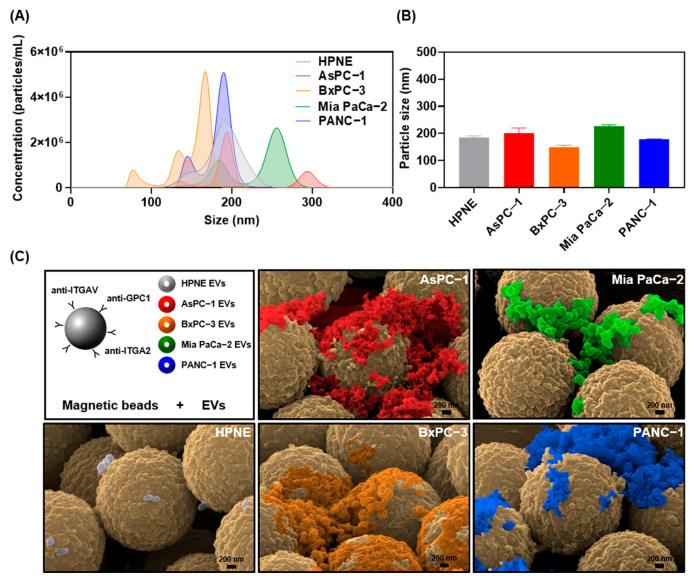
Enrichment of pancreatic tumor cell-derived EVs by immunoprecipitation with magnetic beads. (**A**) Particle size distribution of EVs isolated from HPNE, AsPC−1, BxPC−3, Mia PaCa−2, and PANC−1 cell lines. The area under the curve represents the absolute number of particles isolated from the concentrated culture medium. (**B**) Mean particle sizes of EVs were analyzed in triplicate and graphed. (**C**) SEM images of EVs bound to surfaces of magnetic microbeads after immunoaffinity capture. Microbeads were functionalized with antibodies against the pancreatic cancer associated surface markers ITGAV, ITGA2, and GPC1. Scale bars represent 200 nm.

**Figure 4 ijms-22-13621-f004:**
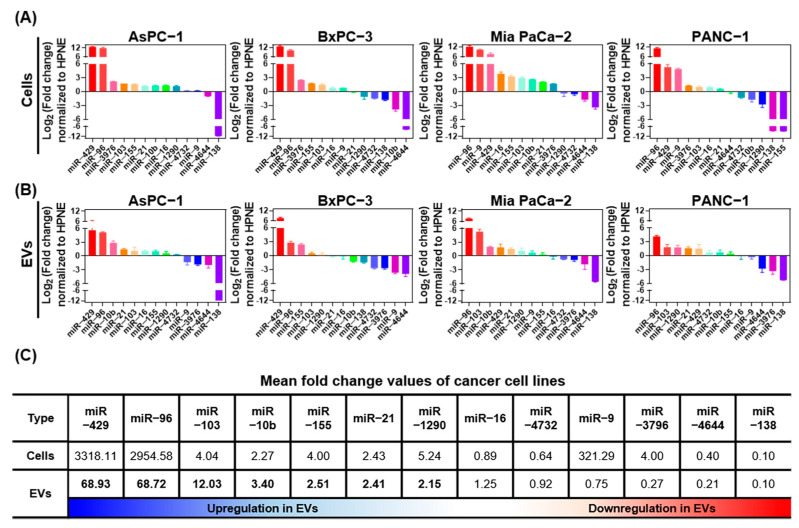
Bar charts showing fold changes in miRNA expression in pancreatic cancer cells and EVs normalized to expression in HPNE cells and EVs. Thirteen candidate miRNAs were analyzed in (**A**) cells and (**B**) EVs by real-time quantitative PCR. (**C**) Mean fold change of miRNA expression in AsPC−1, BxPC−3, Mia PaCa−2, and PANC−1 cancer cells and EVs normalized to expression in HPNE cells and EVs. miRNAs that were upregulated in EVs are shown in bold.

**Figure 5 ijms-22-13621-f005:**
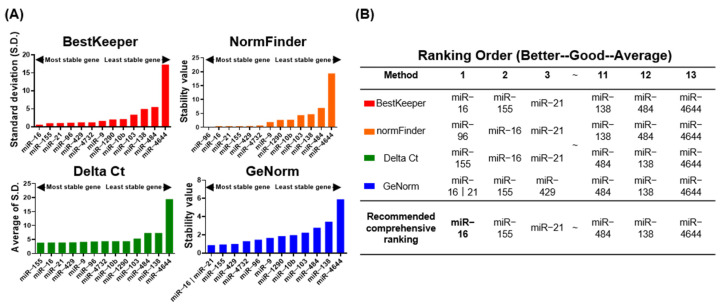
Selection of candidate reference genes using RefFinder. (**A**) Stability of 13 candidate miRNAs in EVs from 40 plasma samples according to Bestkeeper (red), NormFinder (orange), the Delta Ct method (green), and GeNorm (blue) statistical algorithms. (**B**) Rankings generated by RefFinder analysis. The top-ranking reference gene is shown in bold.

**Figure 6 ijms-22-13621-f006:**
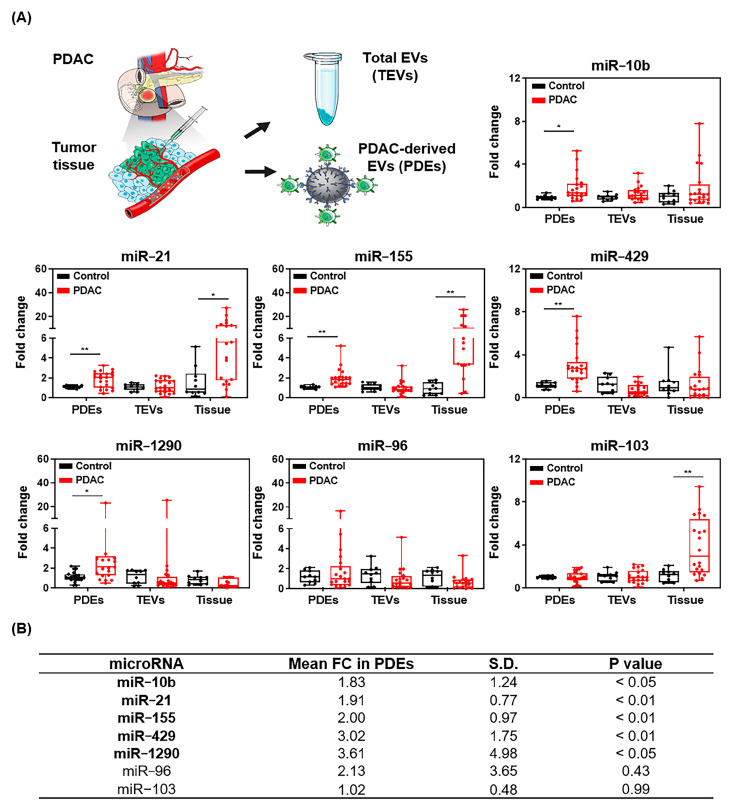
Differential miRNA expression in samples from 20 PDAC patients compared with the normal control. (**A**) Fold changes of candidate miRNA expression in PDEs, TEVs, and tumor tissue. (**B**) Mean fold changes, standard deviations, and P values of candidate miRNAs in PDEs. Upregulated miRNA candidates that were upregulated in PDEs are shown in bold. * *p* < 0.05, ** *p* < 0.01 benign control vs. PDAC in each groups.

**Figure 7 ijms-22-13621-f007:**
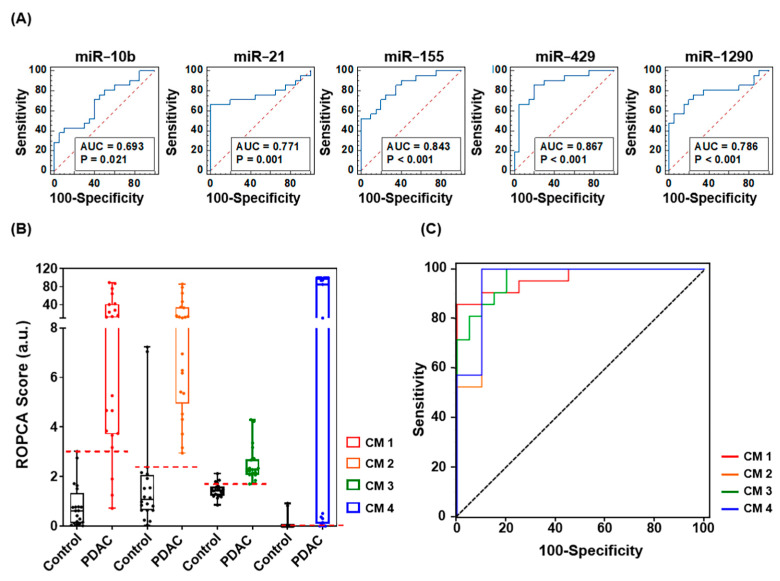
(**A**) ROC curve analyses of the abilities of five candidate miRNAs to discriminate PDAC patients from normal controls. (**B**) ROPCA scoring of CMs 1–4. CM 1 consisted of miR-21, miR-155, miR-429, miR-1290, and CA19-9. CM 2 consisted of miR-10b, miR-21, miR-155, miR-1290, and CA19-9. CM 3 consisted of miR-10b, miR-155, miR-429, miR-1290, and CA19-9. CM 4 consisted of miR-10b, miR-21, miR-155, miR-429, miR-1290, and CA19-9. (**C**) ROC curve analyses of CMs 1–4.

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
