# Peer review of "Tumor-Specific miRNA Signatures in Combination with CA19-9 for Liquid Biopsy-Based Detection of PDAC"

_ijms, 2021, doi:10.3390/ijms222413621_

Round 1

Reviewer 1 Report

The manuscript of Kim MW et al investigates the role of miRNA in extracellular vesicles as potential biomarkers in liquid biopsies for PDAC diagnostics. They established a novel approach for isolation of PDAC derived extracellular vesicles in cell lines and patient samples. Subsequently, the miRNA expression of vesicles (PDE) were correlated with the expression in tumor samples and other EVs, which helped to identify a suitable miRNA biomarker panel with improved sensitivity/specificity compared to established biomarkers (CEA, CA19-9). Overall this is a well presented and technically demanding study with scientific and clinical value. However, major concerns about relevance and validity of this study remain:

Major comments:

  • The patients included in this study, were already diagnosed with PDAC. It seems elusive to investigate biomarker expression and determine their sensitivity/specificity on patients with high tumor burden. Therefore, favorable specificity/sensitivity scores do not reflect their potential as biomarkers for early detection or diagnosis, but only as an additional surveillance marker. The authors may address this by including patients with surgical removal of precursor lesions (IPMNs, MCNs) or other benign pancreatic lesions.
  • A single normal pancreatic cell line is not enough to determine differential marker expression compared to PDE. In addition, normalization of miRNA expression to single normal cell line does not seem convincing. Please consider additional normal pancreatic duct cell lines, such as HPDE for validation of markers.
  • It is not clear why EV markers were selected (ITGA2, ITAGV, GPC1), which appeared to be detected in the tumor cell lines only at low ranges. For all markers analyzed, there appears to be an inconsistent pattern between cell lines and EVs. Please clarify why these markers can be used for PDE isolation, as at this point they do not seem to reliably represent the biologic composition of their parental cells.

Minor comments:

  • In the introduction, the Authors focus too much on the role of biopsies for PDAC diagnosis, which is still greatly irrelevant in majority of PDAC cases. Instead imaging remains the golden standard. Please refer only to the potential of liquid biopsies for early detection in general, and not as potential replacement of tissue biopsies.
  • While the introduction is generally too lengthy, it does not lay down the main points of this study. I suggest rewriting and restructuring the introduction by focusing on the main contribution of the study.
  • The authors should provide more detailed information and clearly note how (criteria) and when the 20 PDAC patients have been selected.
  • The authors should provide more information of which criteria was used to assemble the miRNA multi-panel analyses

Author Response

Dear reviewer,

We, authors appreciate the reviewer’s thoughtful comments made for our manuscript (Manuscript ID: ijms-1488825) entitled "Tumor-specific miRNA signatures in combination with CA19-9 for liquid biopsy-based detection of PDAC”. Based on your comments, we have revised the manuscript accordingly. We very much hope that our revised manuscript should be accepted in the International Journal of Molecular Sciences.

Reviewer 2 Report

  1. Is this article related to “non-invasive biomarkers for early detection before PDAC reaches the incurable stage”, or is it related to “evaluate the combinations for their accuracy in diagnosing PDAC”? These are two different questions? Please clarify. If this is for diagnosis, please provide a rationale as to how this method could be alternative for ERCP guided biopsy and histopathology. What was the AJCC and TNM stages of the 20 PDAC patients? And how do the authors define “early detection” vs “incurable stage”?

  1. What is the basis for selection of “pancreatic cancer-specific markers”?

  1. Please provide data showing that ITGA2, ITGAV, and GPC1 expression are higher only in pancreatic cancer adenocarcinoma patients and not in other adenocarcinomas of the digestive system – such as cholangiocarcinoma, or liver cancers.

  1. This reviewer cannot distinguish between PDEs derived from PDAC cell lines vs PDEs derived from PDAC patients. For instance, in the sentence, “Scanning electron micros-120 copy (SEM) was used to confirm the presence of PDEs along with their morphologies and 121 sizes (Figure 3C)”, are the authors referring to HPNE, AsPC-1, BxPC-3, Mia PaCa-2, and PANC-1 cell lines (lines 119-120) or PDAC patient- derived EVs (PDEs) (Lines 72, 73)? Please use different acronyms for these two different entities and re-provide the manuscript.

Author Response

(The authors gave the same response as above.)
